# Role of Genetic Variations in the Hepatic Handling of Drugs

**DOI:** 10.3390/ijms21082884

**Published:** 2020-04-20

**Authors:** Jose J. G. Marin, Maria A. Serrano, Maria J. Monte, Anabel Sanchez-Martin, Alvaro G. Temprano, Oscar Briz, Marta R. Romero

**Affiliations:** HEVEFARM Group, Center for the Study of Liver and Gastrointestinal Diseases (CIBERehd), Carlos III National Institute of Health, University of Salamanca, IBSAL, 37007 Salamanca, Spain; maserrano@usal.es (M.A.S.); mjmonte@usal.es (M.J.M.); anabelsanchez@usal.es (A.S.-M.); alvarogacho@usal.es (A.G.T.); obriz@usal.es (O.B.); marta.rodriguez@usal.es (M.R.R.)

**Keywords:** detoxification, haplotype, isoform, metabolism, mutation, pharmacogenomics, polymorphism, SNP, transport, variant

## Abstract

The liver plays a pivotal role in drug handling due to its contribution to the processes of detoxification (phases 0 to 3). In addition, the liver is also an essential organ for the mechanism of action of many families of drugs, such as cholesterol-lowering, antidiabetic, antiviral, anticoagulant, and anticancer agents. Accordingly, the presence of genetic variants affecting a high number of genes expressed in hepatocytes has a critical clinical impact. The present review is not an exhaustive list but a general overview of the most relevant variants of genes involved in detoxification phases. The available information highlights the importance of defining the genomic profile responsible for the hepatic handling of drugs in many ways, such as (i) impaired uptake, (ii) enhanced export, (iii) altered metabolism due to decreased activation of prodrugs or enhanced inactivation of active compounds, and (iv) altered molecular targets located in the liver due to genetic changes or activation/downregulation of alternative/compensatory pathways. In conclusion, the advance in this field of modern pharmacology, which allows one to predict the outcome of the treatments and to develop more effective and selective agents able to overcome the lack of effect associated with the existence of some genetic variants, is required to step forward toward a more personalized medicine.

## 1. Introduction

A better understanding of the genetic and molecular bases accounting for the interindividual variability in response to pharmacological treatment requires the knowledge of the role of genetic variants in genes expressed in hepatocytes and involved in the mechanisms affecting hepatic drug handling. These include detoxification mechanisms that are classified into four groups: (i) phase 0, transporters determining the uptake of the drug by hepatocytes; (ii) phase 1, small changes in the molecule induced by oxidation-reduction (redox) reactions; (iii) phase 2, conjugation of drugs with a polyatomic group; and (iv) phase 3, which for some drugs involves the export across the canalicular membrane toward the biliary lumen and, in other cases, involves the transfer across the hepatocyte sinusoidal membrane back to the blood for the elimination by the kidney of the usually more hydrophilic derivatives generated by phases 1/2 in liver cells. The present review is not an exhaustive list but a general overview of the most relevant variants affecting genes involved in detoxification phases because they affect drug activity or induce higher toxicity, in most cases, owing to accumulation in this organ, promoting drug-induced liver injury (DILI) (Table 1). To elaborate this review, we have also taken into account that the liver is the target of many drugs, and hence the existence of genetic variants can affect the efficacy and toxicity of these drugs (Table 2). Additional information to that provided in the present review regarding the clinical relevance of genetic variants, i.e., demonstrated effects on pharmacokinetics, drug response and/or toxicity and their potential clinical impact can be found at databases such as the Pharmacogene Variation Consortium website (https://www.pharmvar.org/gene/), the Clinical Genome (ClinGen) resource (https://clinicalgenome.org/) and the PharmGKB database (http://www.pharmgkb.org/). Minor allele frequency (MAF) data were obtained from the Allele Frequency Aggregator (ALFA) project (version 27 February 2020) (https://www.ncbi.nlm.nih.gov/snp/docs/gsr/alfa/) included in the NCBI database of single nucleotide polymorphisms (dbSNP).

## 2. Phase 0: Uptake of Drugs

During phase 0 of hepatic detoxification, drugs are taken up by the hepatocytes from the blood mainly via carrier-mediated processes that involve members of the solute carrier (SLC) superfamily of proteins. The existence of single nucleotide polymorphisms (SNPs) in these genes, together with the variability of their expression levels in liver cells, are important factors contributing to the interindividual variability in the response of drugs that either act on hepatocytes or whose metabolism and elimination depends on liver function. In many cases, the consequence of genetic variants is the alteration of drug uptake, resulting in a significant impact on its efficacy, safety, and the appearance of adverse reactions. It should be noted that many carriers share overlapping substrate specificity. For this reason, even if the variant reduces the activity of a transporter, this is not always accompanied by essential differences in the clinical response, because it can be compensated by the activity of other transporters.

Databases, such as PharmGKB [1] and the ClinGen resource [2], have included more than 25,000 associations between genetic polymorphisms in *SLC* genes and drug response. Moreover, the International Transporter Consortium (ITC), which is comprised of scientists from academia, industry and regulatory agencies around the world, has documented a high degree of interindividual variability in SLC transporter activities due to the presence of these genetic variants [3].

### 2.1. Organic Anion-Transporting Polypeptides (OATPs)

Some members of the OATP family, such as OATP1B1, OATP1B3, and OATP2B1 (*SLCO1B1*, *SLCO1B3*, *SLCO2B1* genes, respectively) are highly expressed at the basolateral membrane of hepatocytes, where they play an important role in the uptake of many different substrates [4]. Because of their role in drug uptake and disposition, OATP1B1 and OATP1B3 are considered among the most clinically relevant carriers by ITC guidelines [5,6].

OATP1B1 is expressed exclusively in the sinusoidal membrane of hepatocytes. This transporter has a broad substrate specificity that includes anionic, but also zwitterionic and neutral lipophilic compounds. Among them are drugs widely used to reduce the risk of cardiovascular diseases, such as statins, the antihypertensives enalapril, temocapril, olmesartan, and valsartan, and antidiabetics such as repaglinide [7]. OATP1B1 can also transport thiazolidinediones (troglitazone), anticancer drugs (SN-38, methotrexate, and taxanes), antibiotics (rifampicin, benzylpenicillin), antifungals (caspofungin) and immunosuppressants (tacrolimus) [8]. To date, almost 200 SNPs in the *SLCO1B1* gene have been described, some of them very frequent, such as c.388A>G (p.Asn130Asp, rs2306283), whose minor allele frequency (MAF) is 42.8%. This variant has less capacity to transport certain drugs, for instance, repaglinide [9,10]. However, this does not result in an important impact on the pharmacokinetics, response and toxicity of these drugs. In contrast, other *SLCO1B1* variants have considerable clinical importance. This is the case of c.521T>C (p.Val174Ala, rs4149056), which has MAF of 14.7%. When expressed in cells in vitro, this carrier has decreased transport activity, due to diminished expression at the plasma membrane [11] and a higher degree of protein phosphorylation [12]. There are four common haplotypes bearing these two SNPs: *SLCO1B1*1A*, consisting of wild-type sequences (c.388A/c.521T), *SLCO1B1*1B* (c.388G/c.521T), *SLCO1B1*5* (c.388A/c.521C) and *SLCO1B1*15* (c.388G/c.521C). Among them, *SLCO1B1*5* and *SLCO1B1*15* haplotypes have been associated with higher serum concentrations of certain drugs, which may be due to a slower hepatic uptake. Although low clinical impact of these SNPs for many drugs that are substrates of OATP1B1 has been found, they markedly affect the pharmacokinetics of statins [13]. As compared with patients carrying the wild-type *SLCO1B1*1A* haplotype, serum concentrations of statins are higher in patients harboring the c.521T>C variant, which is accompanied by lower drug efficacy together with higher risk of suffering myopathy and rhabdomyolysis [14,15]. The impact of this variant is such that the clinical guidelines released by the ITC and the Dutch Pharmacogenetics Working Group recommend for patients harboring the c.521T>C variant to halve the dose of simvastatin or to replace atorvastatin with fluvastatin, which is a worse OATP1B1 substrate (http://www.pharmgkb.org/) [1]. Moreover, patients carrying this variant have higher plasma levels of the anti-HIV drug atazanavir [16], and it has been proposed to reduce the drug dose when these patients also harbor the variants rs2472677 of *NR1I2* and rs1045642 of *ABCB1* to avoid adverse effects (http://www.pharmgkb.org/) [1].

The substrate specificity of OATP1B3, also highly expressed in hepatocytes, markedly overlaps that of OATP1B1. Like other *SLCO* genes, *SLCO1B3* is polymorphic, and many genetic variants have been associated with reduced transport activity or expression of OATP1B3 in vitro [17]. Among them, the most clinically relevant SNPs are c.334T>G (p.Ser112Ala, rs4149117) and c.699G>A (p.Met233Ile, rs7311358). These two very common SNPs have a frequency of 84.2% and 85.3%, respectively. The presence in the OATP1B3 protein of these two amino acid substitutions, but not when they appear separately, reduces the activity of this transporter expressed in a cellular model [18]. In addition, the presence of both SNPs in patients results in elevated plasma levels of tacrolimus, an immunosuppressant used in patients with renal transplantation [19].

The clinical significance of the presence of genetic variants in other members of the OATP family is not well understood, because only weak evidence or contradictory results have been obtained and, therefore, no recommendations have been made for clinical implementation. Thus, the ITC guidelines do not specifically recommend OATP2B1 research in drug development and personalized medicine. However, OATP2B1 is a drug transporter with a broad substrate specificity expressed in the liver and many other tissues, suggesting that it may have an important, but not yet fully elucidated, role in drug bioavailability. OATP2B1 mediates the uptake of statins, antihistamines (fexofenadine), antidiabetics (glibenclamide, glyburide), antiarrhythmics (amiodarone), and some antitumor drugs (antifolate drugs). Although in vitro studies have shown that *SLCO2B1* SNPs affect the uptake of some of these drugs [20], in vivo confirmation is still required.

### 2.2. Organic Cation Transporters (OCTs)

OCT1 (*SLC22A1*) is the primary hepatic transporter of organic cations. This carrier is highly expressed at the sinusoidal membrane of the hepatocytes, where it mediates the uptake from the blood of a wide variety of cationic drugs [21]. Recent clinical evidence supports the results obtained from in vitro studies regarding the functional impact of genetic variants on OCT1 transport activity and the significant pharmacokinetic changes that accompany these variants.

Many SNPs and *SLC22A1* haplotypes have been linked to clinical effects. Nevertheless, the results are not robust enough for clinical guidelines to include recommendations regarding changes in drug dosage for patients carrying these variants. The five non-synonymous SNPs of *SLC22A1* with clinical consequences that have been most extensively studied are c.181C>T (p.Arg61Cys, rs12208357, MAF 6.9%), c.262T>C/A (p.Cys88Ser/Arg, rs55918055, MAF 0.3%), c.1201G>A (p.Gly401Ser, rs34130495, MAF 2.5%), c.1260_1262del (p.Met420del, rs72552763, MAF 10.4%), and c.1393G>A (p.Gly465Arg, rs34059508, MAF 2.2%). The six haplotypes resulting from these SNPs are *SLC22A1(*1* to **6)* [21]. The Arg61Cys, Cys88Ser/Arg and Gly465Arg variants have a deleterious effect on OCT1 transport capacity, while others like Gly401Ser and Met420del result in a slight reduction in OCT1 function [56]. Patients carrying any of these OCT1 variants showed higher plasma concentrations of metformin [57] together with a lower hypoglycemic response to this drug [58]. In addition, these variants also affect the pharmacokinetics of imatinib, tropisetron, sumatriptan, morphine, tramadol, and sorafenib [56,59]. Other deleterious *SLC22A1* variants have been associated with increased risk of metformin-induced side effects [60]. Although many SNPs have been described in other organic cation transporters, such as OCTN1 (*SLC22A4*) and OCTN2 (*SLC22A5*), these have not been linked to relevant clinical consequences [21].

### 2.3. Organic Anion Transporters (OATs)

Most members of the OAT family have low expression in the liver, except for OAT2 (*SLC22A7*), which is highly expressed in hepatocytes and has broad substrate specificity. Accordingly, the presence of SNPs in *SLC22A7* could be expected to be clinically relevant. However, no robust results have been reported. Patients with colorectal cancer carrying the frequent synonymous variant c.1269C>T (p.Ser423Ser, rs2270860, MAF 33.6%) and the intronic polymorphism g.43272188A>G (rs4149178, MAF 17.3%) have been reported to have a higher risk of capecitabine-induced toxicity [61]. Although several SNPs in the *SLC22A9* gene encoding OAT7 have been described, and some of them reduce pravastatin uptake in vitro, the clinical consequences of these variants are not yet well known [62].

## 3. Phase 1: Drug Oxidoreduction

Several enzymes of the superfamily of cytochrome P450 (CYPs) are involved in the hepatic oxidative metabolism of numerous drugs. Although CYP2C and CYP3A (20%–30% of liver CYPs) are the most important subfamilies, many other CYPs, such as CYP1A2, CYP2B, CYP2D, CYP2E1, CYP2J2, and CYP4F2, are also expressed in the liver and have a clinical impact on drug oxidoreduction [63]. Genetic variants in *CYP* genes play an important role in liver pharmacogenomics. Thus, the presence of multiallelic polymorphisms has been the rationale for classifying individuals as poor (PM), intermediate (IM), extensive (EM) and ultrafast (UM) drug metabolizers, which is closely related to suffering from adverse drug reactions (ADR) and, in some patients, dosage adjustment is required to obtain the desired drug effect.

### 3.1. CYP2B6

This enzyme accounts for the oxidation of ≈3% of drugs metabolized by CYPs, such as efavirenz, bupropion, and cyclophosphamide [64]. This gene is highly polymorphic, with variants affecting efavirenz metabolism and pharmacokinetics, which have become relevant for the treatment of HIV [65]. Patients carrying variants c.516G>T (p.Gln172Gln, rs3745274, MAF 25.9%) and g.21563C>T (rs8192719, MAF 26.3%) suffering from HIV infection may have decreased plasma concentrations and increased clearance of efavirenz as compared to patients with the GT or TT and CT or TT genotype, respectively. Moreover, patients with variants c.516G>T (p.Gln172Gln, rs3745274), c.983T>C (p.Ile328Thr, rs28399499, MAF 0.4%), or both 516G>T/983T>C, as well as haplotypes *CYP2B6*4* and *CYP2B6*6*, have been associated with adverse effects of efavirenz treatment [22].

### 3.2. CYP2C19

There are 30 *CYP2C19* variants, some of which have been widely studied due to their clinical relevance in the treatment of gastric ulcers, depression, thromboembolic disease, and epilepsy. Dosing guidelines for patient management [66] and genotyping platforms have been approved by the Food and Drug Administration (FDA) to detect *CYP2C19* alleles [67]. Thus, variants c.636G>A (p.Trp212Ter, rs4986893, MAF 1.2%) and c.681G>A (p.Pro227Pro, rs4244285, MAF 14.8%), and loss-of-function alleles (**2*, **3*, **4*, **5*, **6*, **7* and **8*) have been associated with a reduction in the efficacy of the antiplatelet agent clopidogrel and with a higher risk of suffering from serious adverse cardiovascular events during treatment with this drug [23]. The treatment with some antibiotics and proton pump inhibitors such as omeprazole, lansoprazole, and pantoprazole, used against *H. pylori* infection, results in higher plasma concentrations and therefore increased efficacy in PM than EM individuals [24] and less effective in UM patients with the **17/*17* genotype [25].

### 3.3. CYP2C8

This enzyme is involved in the metabolism of antidiabetics, non-steroidal anti-inflammatory drugs (NSAIDs), and the anticancer drug paclitaxel [26]. Oral clearance of thiazolidinediones, such as rosiglitazone and pioglitazone, used in the treatment of type 2 diabetes mellitus, is higher in individuals carrying the *CYP2C8*3* allele and variant c.986A>G (p.Ala257Asp, rs10509681, MAF 10.9%). In these patients, the therapeutic effect of these agonists of the peroxisome proliferator-activated receptor (PPAR) is reduced, but also the risk of developing edema, as compared with individuals carrying the variant *CYP2C8*1/*1* [27]. There is conflicting information regarding the relationship between *CYP2C8*3* and the pharmacokinetics of ibuprofen. This may be due in part to the fact that ibuprofen is also a substrate of *CYP2C9*. Concerning adverse effects, some data suggest that the combined presence of *CYP2C8*3* and *CYP2C9*2* plays a determinant role in gastrointestinal bleeding induced by treatment with NSAIDs. However, further studies are needed to confirm this point [68].

### 3.4. CYP2C9

This is the most abundant CYP in hepatocytes (20% of all CYPs). CYP2C9 is also the most important enzyme of the CYP2C subfamily in the metabolism of many drugs, such as warfarin, phenytoin, glipizide, and tolbutamide [69,70]. Two clinically relevant polymorphisms of this gene are **2* and **3*, because all carriers, even the heterozygous ones, with type 2 diabetes mellitus are at risk of hypoglycemia and bleeding during treatment with NSAIDs [63]. Two variants included in these haplotypes c.430C>T (p.Arg144Cys, rs1799853, MAF 11.3%) and c.1075A>G (p.Ile359Leu/p.Ile359Val, rs1057910, MAF 6.8%), respectively, lead to PM phenotypes [71]. Regarding warfarin, these patients require lower doses of this drug to achieve a similar anticoagulant effect to that induced in **1/*1* patients [31]. Lower blood sugar levels during glipizide and tolbutamide therapy have also been reported [32], as well as a higher risk of overdose during phenytoin therapy [33]. Homozygous individuals for variant c.1075A>G (p.Ile359Leu/p.Ile359Val, rs1057910) also have lower clearance of S-acenocoumarol, celecoxib, diclofenac, ibuprofen, nateglinide, fluvastatin, and phenprocoumon than homozygous **1/*1* individuals [34]. Other variants involved in warfarin-induced ADR are **11*, **5*, **6* haplotypes, c.482-2313A>G (rs7089580), c.1080C>A,G (p.Asp360Glu, rs28371686), c.1076T>A/C (p.Ile359Asn/p.Ile359Thr, rs56165452), c.449G>A/C/T (rs7900194), all with MAF <0.01%, and c.820A>C/T (rs4917639, MAF 19.5%). In addition, the pharmacological inhibition of CYP2C9 by drugs such as fluconazole, as well as the existence of genetic variants in genes involved in the pathway accounting for the mechanism of action of warfarin, such as the enzyme vitamin K epoxide reductase complex 1 (VKORC1), influence warfarin clearance.

### 3.5. CYP2D6

Although representing only 1%-5% of liver CYP enzymes, the activity of CYP2D6 accounts for 25%–30% of the metabolism of all prescribed drugs, and 25% of the so-called FDA pharmacogenomic biomarkers. The *CYP2D6* gene is highly polymorphic, and hence there are over 100 allelic variants causing different effects on the activity of this enzyme, which results in different phenotypes [72]. Some of them determine IM phenotypes (**9*, **10*, **17*, **29* and **41*), whereas others account for PM (3 to **8* and **36*) or UM (**1xN*, **2*, and **35*) phenotypes [73]. CYP2D6 metabolizes tamoxifen into endoxyfen that binds to the receptor more efficiently than tamoxifen. Thus, cancer patients carrying allele variants **10*, **3*, **4* (variant c.1846G>A, rs3892097, MAF 18.4%) **41*, **5* and **6* with a PM phenotype showed a greater risk of relapse [35]. The variants mentioned above also have a role in reducing the effect and increasing the risk of secondary effects of some antidepressants such as amitriptyline, clomipramine, desipramine, doxepin, imipramine, nortriptyline and trimipramine. A relationship between the presence of *CYP2D6*10*, **17* (variant c.1023C>T, p.Thr107Ile/Asn, rs28371706, MAF 2.9%), **1xN*, **2*, **2xN*, **3*, **4*, **40*, **41*, **5* and **6* variants and the alteration in the metabolism of codeine to morphine has been reported [37]. Several studies have shown a decrease in morphine levels and in analgesia in PMs that receive codeine compared to IMs, and IMs compared with UMs, which have a higher conversion rate of codeine into morphine, which can lead to toxic systemic concentrations of the later [74]. Other *CYP2D6* variants, such as **3, *4, *5* and **10*, are involved in ADR events during treatment with fluvoxamine, paroxetine, risperidone, flecainide, propafenone, and metoprolol.

### 3.6. CYP3A5

Several members of the CYP3A subfamily, such as CYP3A4, CYP3A5, CYP3A7, and CYP3A43, are responsible for the biotransformation of 40%-50% of the 200 most frequently prescribed drugs [75]. CYP3A5 includes 25 allelic variants, of which *CYP3A5*3* (g.6986A>G, rs776746, MAF 88.2%) is the most studied one, followed by alleles **6* and **7* [76], whereas **3/*3* phenotype implies the absence of expression of this enzyme. Despite the confirmed role of CYP3A5 in the clearance of tacrolimus [38], it has not been consistently associated with the risk of acute rejection after organ transplantation in patients treated with this immunosuppressant. However, there is a dosing equation for the use of tacrolimus in adult kidney transplant recipients, which, together with days post-transplant and other individual factors, includes the *CYP3A5* genotype [39]. Thus, EM and IM individuals generally have decreased dosage adjustment as compared with those who are PMs, possibly due to a delay in achieving optimal blood tacrolimus concentrations. Therefore, EMs and IMs would require a higher recommended starting dose [77].

### 3.7. CYP4F2

This enzyme affects vitamins E and K metabolism. Genetic variants in *CYP4F2* gene altering the bioavailability of vitamin K (e.g., c.1297G>A, p.Val433Met, rs2108622, MAF 28.5%) also have an impact on the dosing of several vitamin K antagonists, such as warfarin and acenocoumarol [40].

## 4. Phase 2: Drug Conjugation

Several enzymes involved in the conjugation with polyatomic moieties play an essential role in drug inactivation by the liver. Moreover, these biotransformations usually result in more polar and hence more water-soluble derivatives, which are more easily secreted into bile or eliminated by the kidney after being extruded from the hepatocyte across the sinusoidal membrane.

### 4.1. Catechol O-methyl Transferases (COMTs)

This group of enzymes is responsible for the transfer of methyl groups from S-adenosyl methionine to catecholamines. The substrates of COMT include endogenous neurotransmitters but also drugs with catechol structures used in the treatment of hypertension, asthma, and Parkinson’s disease [46]. For the soluble form of COMT, there is a polymorphism affecting the amino acid 108 (c.322G>A, p.Val108Met, rs4680) while for the membrane-associated form (with 50 additional N-terminal amino acids), this identical polymorphism (c.472G>A) is at position 158 (p.Val158Met, rs4680). The Met allele (A) is known as the low activity or “L” variant, whereas the Val (G) or “H” allele is the high activity one [46]. Interindividual differences in the cognitive responses to several drugs used in psychiatric illness are related to COMT genetic variants [78]. Moreover, the effect of diuretics, calcium channel blockers, and angiotensin receptors blockers on systolic or diastolic blood pressure is strongly associated with COMT variants. [79]. In addition, before starting the pharmacological treatment of Parkinson’s disease, it is important to consider the gender and the presence of genetic variants in MAO-B and COMT genes (MAO-B-rs1799836 and COMT-rs4680), since the outcome is affected by sexual dimorphism in genes related to dopamine metabolism [80].

### 4.2. Glutathione S-Transferases (GSTs)

GSTs are a family of enzymes able to catalyze the formation of thioether conjugates between glutathione and xenobiotic compounds. Several anticancer drugs, such as platinum derivatives, anthracyclines, *Vinca* alkaloids, cyclophosphamide, and phycotoxins are substrates of GSTs [81]. The family of soluble GST enzymes is divided into eight classes designated Alpha, Kappa, Mu, Pi, Sigma, Theta, Zeta and Omega, among which the most critical genes in liver drug metabolism are *GSTM1, GSTT1, GSTP1,* and *GSTA1*. Common polymorphisms for *GSTP1* have been described, which are associated with the toxicity and response to several anticancer drugs and, in particular, platinum-related agents [82]. Thus, the non-synonymous polymorphism of *GSTP1* p.Ile105Val (c.313A>G in exon 5, rs1695), sometimes referred to as *GSTP1*B*, is considered as a risk factor in the early onset of neurotoxicity during the treatment of colorectal cancer with oxaliplatin [42].

### 4.3. N-acetyl Transferases (NATs)

These are cytosolic enzymes found in numerous tissues of different species [46]. In the adult human liver, arylamine-N-acetyltransferases are involved in the biotransformation of aromatic amines and hydrazines [46]. NAT1 and NAT2 are polymorphic enzymes (28 alleles for *NAT1* and 88 alleles for *NAT2*) [83]), and such polymorphisms determine the acetylator phenotype. NAT1 activity may have implications in the response to anticancer drugs. Thus, *NAT1* polymorphisms that produce changes in the enzymatic activity may affect the response of these individuals to chemotherapy. *NAT1*4* and *NAT2*4* are the reference or wild type alleles for the respective genes [84,85]. In contrast to *NAT1*, *NAT2* has a high frequency of functional variation and diversity of haplotypes that differ among ethnic groups. *NAT2* genotypes can be grouped into three different phenotypes “slow acetylator” (two slow alleles), “intermediate acetylator” (one slow and one fast one) and “fast acetylator” (two fast alleles) [84].

### 4.4. Sulfotransferases (SULTs)

These enzymes catalyze the conjugation of 3′-phosphoadenosine 5′-phosphosulfate (PAPS) with an acceptor group (N-, O- or S) in SULT substrates [46]. Sulfonation has an essential role in the biotransformation of many endogenous and xenobiotic compounds. Four families of SULTs (SULT1, SULT2, SULT4, and SULT6) with at least 13 different members have been described. SULTs are divided into membrane-linked and cytosolic enzymes [46]. The best known genetic polymorphisms are those of the hepatic isoform SULT1A1 [46]. The most common allelic variant of *SULT1A1* gene in different populations is c.638G>A (p.Arg213His, rs9282861), known as *SULT1A1*2,* which is associated with higher thermolability and lower enzymatic activity as compared to the wild-type variant [45]. *SULT1A1*3*, SNP 667A>G (p. Met223Val, rs1801030), is the second most frequent polymorphism, which generates an enzyme with a higher affinity for PAPS and SULT substrates [86].

### 4.5. Thiopurine-S-Methyltransferase (TPMT)

TPMT activity depends on S-adenosyl methionine to catalyze the S-methylation of aromatic and heterocyclic sulfhydryl compounds that include anticancer and immunosuppressive thiopurines, such as 6-mercaptopurine (6-MP), 6 thioguanine (6-TG) and azathioprine, used for the treatment of acute lymphoblastic leukemia, autoimmune disorders, inflammatory bowel disease, and recipients of solid organ transplantation. The altered enzymatic activity of TPMT causes the accumulation of thiopurine nucleotides and the cytotoxic symptoms leading to hematopoiesis failure [46]. TPMT is a cytosolic enzyme with high expression in the liver and kidney, where its overall activity is determined by the existence of genetic polymorphisms [46]. Up to 28 different alleles of the *TPMT* gene have been identified [87] and associated with great interindividual variability in the therapeutic efficacy and toxicity of thiopurines. Among them, the molecular and clinical implications of *TPMT*2, TPMT*3A, TPMT*3B,* and *TPMT*3C* have been well studied. The variant *TPMT*2* is p.Ala80Pro (c.G238>C, rs1800462) [88], which leads to a protein with 100 times reduced catalytic activity. When patients carrying this allele are treated with standard doses of thiopurines, excessively high levels of these drugs may be reached and hence they have a higher risk of side-effects due to myelosuppression [88]. The variant *TPMT*3C*, p.Tyr240Cys (c.719A>G, rs1142345) is the most common allele in African-American and East Asian populations. *TPMT*3C* is associated with decreased TPMT enzymatic activity due to enhanced protein degradation [89]. *TPMT*4* (rs1800584) includes a transition from G>A in the final splice acceptor nucleotide in intron 9 of the *TPMT* gene. The presence of this variant results in deficient TPMT activity in carrier subjects [90]. *TPMT*3A* haplotype contains two non-synonymous SNPs, **3B* and **3C* and is the most common haplotype in the Caucasian population. The *TPMT*3A* variant results in a significant decrease in TPMT enzymatic activity leading to a higher risk of toxicity when a thiopurine-based therapy is administered [90].

### 4.6. UDP-Glucuronyltransferases (UGTs)

This is a superfamily of membrane-bound enzymes that catalyze the formation of chemical bonds between nucleophilic O, N, S, or C atoms with uridin-5′-diphosphate-ß-D-glucuronic acid [46]. Among the 22 human UGT proteins, UGT1A1, 1A3, 1A4, 1A6, 1A9, 2B7, and 2B15 isoforms are considered the most important in drug metabolism [46]. Liver UGT1A1 is the only enzyme responsible for bilirubin metabolism [91]. The high number of UGT1A1 variants includes enzymes with reduced or increased activity, which results in an altered phenotype [91]. Thus, the *UGT1A1*6* allele p.Gly71Arg (c.211G>A, rs4148323) has been associated with hyperbilirubinemia during the treatment with the anti-HIV drug indinavir [92]. The *UGT1A1*28* allele (rs3064744) consists of seven TA repeats in the promoter region, which leads to ≈70% decrease in its transcriptional activity and hence diminished gene expression [93,94]. Both *UGT1A1*6* and *UGT1A1*28* alleles have been extensively studied with regard to drug toxicity in particular to irinotecan, which can produce harmful side effects such as neutropenia and diarrhea [47]. Although the *UGT1*28* allele has also been associated with a higher risk of irinotecan-induced side effects, other studies have failed to confirm these noxious effects of irinotecan in carriers of either **28* or **6* genotypes [95,96,97]. Thus, validation trials to elucidate the usefulness of genotyping these alleles as a convenient practice before treating colorectal cancer patients with irinotecan are still needed.

## 5. Phase 3: Drug Export

Among detoxification processes, phase 3 consists of the extrusion of xenobiotics from the hepatocytes mainly through ATP-binding cassette (ABC) export pumps [8]. Most genes of this superfamily are highly polymorphic, which has an impact on the functionality of these transporters, resulting in increased or decreased drug concentrations in hepatocytes. These changes can alter the pharmacokinetics of their substrates, affecting the efficacy and/or toxicity of these drugs [98]. Thus, according to data available at the PharmGKB database [1], the presence of SNPs in *ABC* genes, such as *ABCA1, ABCB1, ABCB5, ABCC1-7, ABCC10, ABCG1,* and *ABCG2*, has been associated with altered drug response and toxicity. However, high or moderate clinical relevance (levels 1 to 2 in the PharmGKB database classification) [1] of genetic variants has been reported only in the case of *ABCB1, ABCC4, ABCC7,* and *ABCG2* [98].

### 5.1. Multidrug Resistance Protein 1 (MDR1)

MDR1, also called P-glycoprotein (P-gp, gene *ABCB1*), is expressed in many epithelial cells, where it accounts for the transport of a wide variety of drugs [99]. MDR1 is highly expressed at the canalicular membrane of hepatocytes, where this pump mediates the secretion into bile of endogenous and xenobiotic compounds. There is significant interindividual variability in the function of this drug export pump, which may be due to the presence of numerous genetic variants, whose effect on the transport of hundreds of drugs has been studied in vitro. However, according to the PharmGKB classification, only two SNPs in the *ABCB1* gene have clinical relevance (level 2): c.2677T>G/A (p.Ser893Ala, rs2032582, MAF 61.7% and 4.9% for G and A alleles, respectively) and c.3435T>C (p.Ile1145Ile, rs1045642, MAF 48.8%) [100]. The presence of both variants is accompanied by high MDR1 expression [101]. In addition, rs2032582 variant has been associated with decreased simvastatin concentrations in hepatocytes, probably due to an increase in MDR1-mediated simvastatin secretion into the bile. Patients carrying this variant experience lower simvastatin-induced cholesterol reduction compared to patients with the wild-type genotype [48]. Similar MDR1-mediated enhanced hepatic clearance accompanying these SNPs has also been reported to affect the efficacy of other drugs, such as ondansetron [49] and fentanyl [50]. Moreover, individuals carrying the variant rs1045642 are less likely to suffer from methotrexate-induced liver toxicity than those with the wild-type variant [51]. The presence of the rs1045642 variant also decreases the risk of hepatic toxicity induced by other drugs such as digoxin [52]. In contrast, in children with acute lymphoblastic leukemia carrying this variant, a relationship between lower plasma levels of methotrexate and a higher risk of neutropenia and thrombocytopenia has been found [102].

### 5.2. Breast Cancer Resistance Protein (BCRP)

BCRP (gene *ABCG2*) is also expressed in the canalicular membrane of hepatocytes, where this pump participates in the extrusion into bile of a wide variety of drugs, with a substrate specificity that partially overlaps that of MDR1 [103]. Besides more than 100 intronic variants that lead to reduced mRNA expression, many SNPs have been identified in the coding region of the *ABCG2* gene [104]. Two common polymorphisms are c.34G>A (p.Val12Met, rs2231137, MAF 7.0%) and c.421C>A (p.Gln141Lys, rs2231142, MAF 10.1%). When the variant Val12Met was expressed in K562 cells, this mutation reduced the capacity of BCRP to transport tyrosine kinase inhibitors [105]. However, clinical studies have not found any relationship between the presence of this variant and the pharmacokinetics of BCRP substrates [103]. The variant rs2231142 affects the nucleotide-binding domain of the transporter and decreases protein levels because it promotes its degradation. It has been recommended that patients with gout carrying the rs223114 variant in *ABCG2* take a higher dose of allopurinol than patients with the wild-type genotype [53]. However, this recommendation should be reviewed, because the dosage of the drug is based on urate plasma levels, which as a BCRP substrate can be influenced by changes in the expression and activity of this pump. It has also been shown that patients taking rosuvastatin who carry the allele rs2231142 in *ABCG2* have higher plasma drug concentrations, probably due to lower secretion of this drug into bile [54]. Higher accumulation of rosuvastatin in the liver enhances its pharmacological effect, resulting in a more marked reduction in cholesterol levels, but also a higher risk of toxicity [106]. When this variant is present together with rs4149056 in the *SLCO1B1* gene, a limitation of the dose of rosuvastatin and atorvastatin is recommended to prevent side effects [107]. It has been suggested that individuals carrying this variant could also have increased risk of suffering from drug-induced toxicity during treatment with sulfasalazine, sunitinib, gefitinib, and other statins (simvastatin and fluvastatin) [108]. Accordingly, the ITC considers this polymorphism as clinically relevant [108].

### 5.3. Multidrug-Resistance Associated Protein 4 (MRP4)

MRP4 (gene *ABCC4*) is expressed at the basolateral membrane of the hepatocytes, where it plays an important role in preventing the accumulation of bile acids by exporting them to the blood under cholestatic conditions [109]. MRP4 has broad substrate specificity, including among its substrates, nucleoside analogs and antiviral drugs. Some SNPs of this highly polymorphic gene have clinical impact [109]. For instance, when treated against HIV with tenofovir, patients carrying the very frequent SNP c.3348G>A (p.Lys1116Asn, rs1751034, frequency of 80.2%) in the *ABCC4* gene have lower plasma levels of this drug, due to its enhanced MRP4-mediated liver and kidney clearance [55].

### 5.4. Other ABC Genes

The presence of the variants c.656G>A (p.Arg219Lys, rs2230806) and g.49960C>G (rs12003906) in *ABCA1* gene results in a decreased function and expression, respectively, of the ABCA1 transporter, which leads to an increased hepatic accumulation of hypolipemic drugs, such as pravastatin and fenofibrate, and hence an enhanced response to these drugs [110]. However, the ability of ABCA1 to transport these drugs has not been demonstrated. Even though more than 100 SNPs have been described in other *ABC* genes, especially in those of the subfamily ABCC, and for many of them in vitro studies have shown an impact on the expression and function of these transporters, conflicting results have been obtained in clinical studies regarding the relationship between the presence of these SNPs and changes in drug response or induced toxicity [111].

## 6. The Liver as the Target Organ for Drugs

To carry out their therapeutic effect, several important families of drugs act upon targets located in liver cells and the existence of genetic variants affecting the genes encoding these targets may, therefore, alter the individual response to these drugs.

### 6.1. Cholesterol-Lowering Drugs

Statins are the most widely used lipid-lowering drugs due to their efficacy in the management of dyslipidemias. Among the genes involved in their mechanism of action are: *HMGCR,* which encodes 3-hydroxy-3-methylglutaryl-CoA reductase (HMG-CoA reductase); *APOE*, which encodes apolipoprotein E involved in the transport of cholesterol through the bloodstream; *LDLR*, which encodes the LDL receptor, and some others. The existence of genetic variants in these genes has important clinical implications on the effects of statins [135,136].

Statins are competitive inhibitors of HMG-CoA reductase, the rate-limiting enzyme of the biosynthesis of cholesterol by hepatocytes. There are two frequent and tightly linked intronic SNPs in the *HMGCR* gene: SNP12 g.74642855A>T (rs17244841) and SNP29 g.74655498T>G (rs17238540). Although their effects on the expression and activity of this enzyme are unknown, they have been associated with a lower efficacy of pravastatin therapy regarding the decrease in both total cholesterol and LDL-cholesterol [112]. Genotypes CT and TT of the g.74615021C>T (rs17671591) variant have been associated with increased response to atorvastatin (greater reduction in LDL-cholesterol) in patients with hypercholesterolemia, as compared to genotype CC [113].

Conflicting results have been reported by studies assessing the effect of APOE variants on the lipid-lowering efficacy of statins [136]. This gene is expressed as one of the three alleles, which, in order of increasing affinity for the LDL receptor are: E2, E3 (wild-type) and E4. Carriers of the *APOE* E2 allele (c.526C>T, p.Arg176Cys, rs7412) have a greater reduction of total cholesterol, LDL-cholesterol and triglycerides levels when treated with simvastatin [114], pravastatin [115] or atorvastatin [116]. Some studies have reported that lovastatin therapy was less effective in people carrying the *APOE E4* allele (c.388T>C, p.Cys130Arg, rs429358) [137]. However, other authors have not found any impact of these variants on the response to this drug [138]. The c.-491A>T (rs449647) polymorphism produces a significant decrease in *APOE* promoter activity as the result of altered interactions with transcription factors. This can also influence the response to statin treatment, as carriers of the T allele treated with atorvastatin showed a significantly higher reduction in LDL-cholesterol levels than carriers of the wild-type allele [117]. On the contrary, T carriers present a less satisfactory response to the lipid-lowering drug bezafibrate [117].

Genetic variants in the *LDLR* locus can affect lipid homeostasis, cardiovascular disease risk, and drug response to statin treatment. Thus, patients with vascular diseases carrying the GG genotype of the genetic variant g.11242765A>G (rs2738466) or the TT genotype of the genetic variant g.11242658T>C (rs1433099) have a better response to pravastatin as compared with those carrying the more frequent AA and CC genotypes, respectively [118].

Cholesteryl ester transfer protein (CETP) is involved in the transport of cholesteryl esters to the liver. The g.57005479C>A (rs1532624) polymorphism on the *CETP* gene has been associated with the response to statin therapy in a cohort of elderly patients. Thus, individuals with hyperlipidemia and AA and AC genotypes had a poorer response to HMG-CoA reductase inhibitors than patients with CC genotype [119].

The *CYP7A1* gene encodes cholesterol 7-hydroxylase, the rate-limiting enzyme in the neutral pathway of bile acids synthesis from cholesterol. Patients with the GG and GT genotypes of the SNP c.-267C>A (rs3808607) have a decreased response to atorvastatin as compared to patients with the TT genotype. However, contradictory findings have been reported, which prevents these genetic data from this gene from influencing clinical decisions on statin administration [116,139].

### 6.2. Antidiabetic Drugs

Oral antidiabetic drugs used for the treatment of type 2 diabetes mellitus belong to different families of compounds (biguanides, sulfonylureas, thiazolidinediones, meglitinides, and others). There is a considerable amount of information on the role of genetic variants in the response to diabetic medications [140,141,142]. Although the main targets of many oral antidiabetic drugs are located in the pancreas and adipose tissue, to fully accomplish their therapeutic effect, the mechanisms of action of some of these drugs also involve targets located in hepatocytes.

Ataxia-telangiectasia mutated (ATM) serine/threonine kinase, known to be involved in DNA repair and cell cycle control, plays a role in the effect of metformin upstream of AMP-activated protein kinase. The C allele of the frequent (57%) variant g.108283161C>A (rs11212617) has been associated with a better response to metformin of type 2 diabetes mellitus patients as compared with individuals carrying the A allele [120,121].

In the liver, insulin receptor substrate 1 (IRS1) and IRS2 are essential in insulin-dependent regulation of glucose and lipid metabolism. When the role of *IRS1* variants in the failure of treatment with oral antidiabetic drugs (mostly sulfonylureas and metformin) was investigated, a significant association with the loss-of-function SNP c.2911G>A (p.Gly971Arg, rs1801278) was found [122]. Thus, treatment with oral antidiabetic drugs has >80% higher risk of failing in homozygous individuals for the A allele than in homozygous G carriers [122].

Thiazolidinediones are a therapeutic option for patients with type 2 diabetes mellitus by improving their sensitivity to insulin and β-cell secretory function. The target of thiazolidinediones is PPAR gamma (*PPARG* gene), a nuclear receptor acting as a lipid sensor. The most frequent variant of *PPARG* is c.34C>G (p.Pro12Ala, rs1801282, MAF 9.9%), which has been associated with a better response to pioglitazone and has been proposed as a biomarker for identifying patients more likely to respond to this drug [123].

### 6.3. Antiviral Drugs

Most drugs used to treat viral hepatitis act on the replicative machinery of the virus. However, SNPs in *IMPDH2* gene encoding the enzyme inosine 5′-monophosphate dehydrogenase type II have been found to affect the response of patients receiving antiviral hepatitis drugs, such as ribavirin, taribavirin, merimepodib, and mycophenolate mofetil [143,144]. This gene has low genetic diversity, but the non-synonymous variant c.787C>T (p.Leu263Phe, rs121434586) has a significant impact on IMPDH2 activity, which can hence contribute to the interindividual variability regarding the antiviral efficacy of ribavirin [145].

### 6.4. Anticoagulant Drugs

Oral anticoagulants, such as warfarin, acenocoumarol, phenprocoumon, and phenindione are structurally similar to vitamin K and act as competitive inhibitors of the enzyme VKORC1. However, many patients do not satisfactorily respond to these drugs. In the case of warfarin, differences in the genotype contribute to the interindividual variability in the dose required to reach therapeutic anticoagulant effects [146,147]. The presence of four frequent and tightly linked SNPs located at the *VKORC1* gene regulatory regions, i.e., c.-1639C>T (rs9923231), at the promoter, and c.173+1000G>A (rs9934438), c.173+1369C>G (rs8050894) and c.173+324A>C (rs2884737), at intron 1 [124,148], increases the expression of the enzyme and thus the response to warfarin. These SNPs are very frequent, with MAF ranging from 21% to 38%, although this varies according to ethnic factors [149]. Homozygous and heterozygous genotypes of these SNPs have been associated with high or intermediate sensitivity to warfarin, respectively. Accordingly, in c.-1639C>T carriers, it has been recommended to adjust warfarin dosage. Thus, to effectively inhibit clotting factor activation, a higher warfarin dose is needed in patients with CC genotype than in TC genotype carriers, whereas a lower dose is needed in individuals with TT genotype. In support of this concept, patients carrying this SNP have increased risk of bleeding when treated with warfarin [125].

Other VKORC1 SNPs, such as c.106G>T (p.Asp36Tyr, rs61742245), c.173+525G>A (rs17708472, at intron 1), c.174-1133A>G (rs2359612, at intron 2), and c.*134C>T (rs7294, at 3′-UTR), are associated with lower sensitivity to coumarin derivatives due to decreased expression of this enzyme. Accordingly, to achieve therapeutic effects in patients carrying these SNPs, an increase in the dose of these drugs has been recommended [126,127].

Non-synonymous variants in the *VKORC1* coding region such as c.85G>T (p.Val29Leu, rs104894539), c.134T>C (p.Val45Ala, rs104894540), c.172A>G (p.Arg58Gly, rs104894541) and c.383T>G (p.Leu128Arg, rs104894542) also contribute to the differential response to warfarin and coumarin derivatives [150,151].

### 6.5. Anticancer Drugs

Although a few patients with hepatocellular carcinoma (HCC) can benefit from chemotherapy, many others do not respond. In that respect, several SNPs in a variety of genes belonging to target pathways of anticancer drugs can predict the lack of response to pharmacological treatment of HCC. For instance, SNPs in genes of the vascular endothelial growth factor (VEGF) signaling pathway have clinical relevance in patients receiving sorafenib. Thus, the missense variant c.1416A>T (p.Gln472His, rs1870377, MAF 22.1%) in the *KDR* gene (also known as *VEGFR2*) affects the fifth NH_2_-terminal Ig-like domain within the extracellular region of the receptor and has been reported to decrease the ability of KDR to bind VEGF [152]. In a Chinese HCC cohort, homozygous patients for the wild-type allele (AA) of this variant showed better response to sorafenib and longer time to progression than patients with heterozygous (TA) or homozygous (TT) genotype [128]. This SNP has also been associated with a higher risk of adverse reactions to sorafenib, which include hypertension and hand-foot skin reaction [129]. The c.889G>A (p.Val297Ile, rs2305948) variant, which affects the third NH_2_-terminal Ig-like domain within the extracellular region of the VEGFR-2 receptor, also has an impact on the ligand binding efficiency [152]. The AA genotype of this variant was associated with a longer time to progression [128]. The promoter variant c.-604T>C (rs2071559) alters the binding site for the transcription factor E2F, resulting in a decreased *KDR* expression [152]. A homozygous genotype for the C allele has been associated with shorter overall survival in HCC patients treated with sorafenib [128].

A relationship between the response of HCC patients to sorafenib and the presence of SNPs in *VEGF* genes, which encode ligands of VEGF receptors, has been reported. Thus, the combination of GG genotype of the SNP c.-94C>G (rs2010963) at the 5′-UTR region of the *VEGFA* and CC genotype of the intronic SNP c.1422+200T>C (g.176687621T>C, rs4604006) in *VEGFC* was associated with worse outcomes in HCC patients receiving sorafenib. However, it remains unclear whether those SNPs are associated with changes in circulating levels of VEGF [130]. Although there is no information on the relationship between the presence of *VEGF* variants and the response of HCC to other targeted therapies, these variants have been related to regorafenib response in patients with metastatic colorectal cancer [153]. The results from genotyping angiogenesis-related genes show that only *VEGFA* variant rs2010963 maintains an independent correlation with progression-free and overall survival in these patients [153], which has led researchers to propose *VEGFA* rs2010963 genotyping as a useful tool for a more accurate selection of optimal candidates for regorafenib therapy among metastatic colorectal cancer patients [153].

Polymorphisms in other genes directly related to the VEGF-dependent angiogenesis pathway, such as c.-813C>T (g.150992991C>T, rs2070744) in *eNOS3*, c.714G>T (p.Thr238Thr, rs55633437) in *ANGPT2* and c.1065+2924G>A (g.61730580G>A, rs12434438) in *HIF1A* have also been linked to sorafenib efficacy in HCC patients [131,154]. In these studies, homozygous for the *eNOS3* SNP and patients carrying at least one T allele of the *ANGPT2* SNP had worse overall and progression-free survival [131,154]. Another element that plays a key role in angiogenesis is the hypoxia-inducible factor 1-alpha, a transcription factor encoded by the *HIF1A* gene and involved in VEGF expression. The presence of the A allele of the *HIF1A* rs12434438 variant has been related to a better outcome in patients receiving sorafenib. Accordingly, this variant has been proposed as an independent predictor for a favorable response of HCC to sorafenib [131,154].

Regarding cholangiocarcinoma (CCA), several variants have been associated with the efficacy of drugs commonly used in the treatment of this cancer. The combination of two single polymorphisms, c.354T>C (p.Asn118Asn, rs11615) in *ERCC1*, a DNA repair gene, and c.136-14361C>A (g.113241753C>A, rs12686377) in *SLC31A1*, encoding CTR1, one of the main transporters of platinum derivatives, has been proposed as a potential biomarker to predict the response to the combined treatment of gemcitabine with cisplatin, which is the first-line chemotherapeutic regimen for CCA patients [133]. In addition, a higher expression of the truncated ΔN isoform Δ133p53 of the *TP53* gene detected in tumor tissues has been associated with worse outcomes in CCA patients [155], but no clinical correlation with drug response has been reported. A relationship between high levels of that p53 variant and the resistance to 5-fluorouracil, the second-line therapy for CCA, has been confirmed by an in vitro study [156].

Since conventional therapy for CCA has limited effectiveness, new strategies based on the tumor mutational profile are being studied. The investigation of targeted therapies to treat this cancer has been focused on somatic mutations frequently occurring in CCAs, such as the gain-of-function mutations c.394C>T/A/G (p.Arg132Cys/Ser/Gly, rs121913499) and c.395G>T (p.Arg132Leu, rs121913500) in the *IDH1* gene and several *FGFR2* fusions or translocations [157]. A recent phase III study with ivosidenib, an inhibitor of the mutant IDH1 protein, has shown improved progression-free survival in CCA patients with mutated *IDH1* [158]. The results of a phase II trial for pemigatinib, an FGFR inhibitor, in CCA patients harboring *FGFR2* fusions or rearrangements are also promising [159].

In addition to the alterations in its intracellular targets, sorafenib requires crossing the plasma membrane to reach them. This is hampered by the reduced function of OCT1, which frequently occurs in HCC and CCA cells. Among the reasons accounting for this loss-of-function, which have recently been elucidated [160,161], is the existence of up to seven inactivating somatic mutations identified both in HCC and CCA [56]. In a series of 23 HCC and 15 CCA, the probability of containing at least one of these mutations was 48% in HCC and 40% in CCA [56].

## 7. Conclusions and Perspectives

The existence of genetic variants affecting a high number of genes expressed in liver cells or mutations generated in liver tumors has important clinical repercussions due to the role of this organ in drug handling but also because, for many drugs, the liver is the target for their mechanism of action. The present review has highlighted the importance of defining the genomic profile concerning drug-related genes to predict the outcome of treatments based on these drugs as well as to develop more efficient and selective agents able to overcome the lack of effect associated with the existence of genetic variants, which may affect the hepatic handling of drugs in many ways, such as (i) impaired uptake, (ii) enhanced export, (iii) altered metabolism due to decreased activation of prodrugs or enhanced inactivation of active compounds, and (iv) altered molecular targets located in the liver due to genetic changes or activation/downregulation of alternative/compensatory pathways. In conclusion, the advancement in this field of modern pharmacology, which allows one to predict the outcome of the treatments and to develop more effective and selective agents able to overcome the lack of effect associated with the existence of some genetic variants, is required to bring forward a more personalized medicine.

## Figures and Tables

**Table 1 ijms-21-02884-t001:** Genetic variants affecting genes involved in mechanisms of liver drug handling.

Phase	Gene	Variant	Drugs Affected	Consequences	References
0	*SLCO1B1*	rs4149056	Statins, Atazanavir	Lower liver uptake.	[13,16]
				Enhanced toxicity	
			Pravastatin, Rosuvastatin	Lower liver uptake.	[14,15]
				Reduced efficacy	
1	*CYP2B6*	rs3745274	Efavirenz	Decreased plasma concentrations	[21,22]
		rs8192719			
		rs28399499			
1	*CYP2C19*	rs4986893	Clopidogrel	Lower activity	[23]
		rs4244285	Omeprazole, Lansoprazole	Higher plasma concentrations	[24,25]
1	*CYP2C8*	rs1050968	Paclitaxel, Rosiglitazone, Pioglitazone	Lower response	[26,27,28,29,30]
1	*CYP2C9*	rs1799853	Warfarin, Phenytoin, Glipizide, Tolbutamide	Higher response. Enhanced toxicity	[31,32,33,34]
		rs1057910			
		rs7089580			
		rs28371686			
		rs56165452			
		rs4917639			
		rs7900194			
1	*CYP2D6*	rs3892097	Tamoxifen, Codeine	Lower response.	[35,36,37]
		rs28371706		Enhanced toxicity	
1	*CYP3A5*	rs776746	Tacrolimus	Lower response	[38,39]
1	*CYP4F2*	rs2108622	Warfarin, Acenocoumarol	Lower response	[40]
2	*COMT*	rs4680	Catechol-related drugs	Enhanced toxicity	[41]
2	*GSTP1*	rs1695	Oxaliplatin	Enhanced toxicity	[42]
2	*NAT1*	rs13253389	Cotinine	Lower response.	[43]
				Enhanced toxicity	
2	*NAT2*	Several *	Isoniazid, Pyrazinamide, Rifampicin	Lower response.	[44]
				Enhanced toxicity	
2	*SULT1A1*	rs9282861	SN-38, Flavopiridol, Raloxifene, Ezetimibe	Lower response	[45]
		rs1801030			
2	*TPMT*	rs1800462	Thiopurine drugs (6-Mercaptopurine, 6 Thioguanine, Azathioprine)	Enhanced toxicity	[46]
		rs1142345			
		rs1800584			
2	*UGT1A1*	rs4148323	Indinavir, Irinotecan	Enhanced toxicity	[47]
		rs8175347			
3	*ABCB1*	rs2032582	Simvastatin, Ondansetron	Higher hepatic clearance.	[48,49]
				Reduced efficacy	
		rs1045642	Ondansetron, Fentanyl	Higher hepatic clearance.	[49,50]
				Reduced efficacy	
			Methotrexate, Digoxin	Higher hepatic clearance.	[51,52]
				Lower toxicity	
3	*ABCG2*	rs2231142	Allopurinol	Increased plasma urate.	[53]
				Higher dose recommended	
			Rosuvastatin	Lower hepatic clearance.	[54]
				Higher efficacy and toxicity	
3	*ABCC4*	rs1751034	Tenofovir	Lower plasma levels.	[55]
				Reduced efficacy	

Only variants for which there is evidence of high or moderate clinical relevance (levels 1 to 2 in the PharmGKB database classification) have been included. *, See https://www.pharmgkb.org/gene/PA18/clinicalAnnotation.

**Table 2 ijms-21-02884-t002:** Genetic variants affecting genes involved in the liver response to drugs.

Group	Gene	Variants	Drugs Affected	Consequences	References
Cholesterol lowering drugs	*HMGCR*	rs17244841	Pravastatin	Reduced response	[112]
		rs17238540			
		rs17671591	Atorvastatin	Increased response	[113]
	*APOE*	rs7412	Simvastatin, Pravastatin, Atorvastatin	Increased response	[114,115,116]
		rs449647	Lovastatin, Bezafibrate	Altered response	[117]
	*LDLR*	rs2738466	Pravastatin	Increased response	[118]
		rs1433099			
	*CETP*	rs1532624	Statins	Reduced response	[119]
Antidiabetic drugs	*ATM*	rs11212617	Metformin	Reduced response	[120,121]
	*IRS1*	rs1801278	Sulfonylureas, Metformin	Reduced response	[122]
	*PPARG*	rs1801282	Pioglitazone	Increased response	[123]
Anticoagulant drugs	*VKORC1*	rs9923231	Warfarin	Increased response Toxicity	[124,125]
		rs9934438			
		rs8050894			
		rs2884737			
		rs7294	Warfarin	Reduced response	[126,127]
		rs61742245			
		rs2359612			
		rs17708472			
Anticancer drugs	*KDR*	rs1870377	Sorafenib	Reduced response Toxicity	[128,129]
		rs2305948	Sorafenib	Increased response	[128]
		rs2071559	Sorafenib	Reduced response	[128]
	*VEGFA*	rs2010963	Sorafenib	Reduced response	[130]
	*VEGFC*	rs4604006	Sorafenib	Reduced response	[130]
	*eNOS3*	rs2070744	Sorafenib	Reduced response	[131]
	*ANGPT2*	rs55633437	Sorafenib	Reduced response	[131]
	*HIF1A*	rs12434438	Sorafenib	Increased response	[132]
	*ERCC1*	rs11615	Gemcitabine, Cisplatin	Increased response	[133]
	*SLC31A1*	rs12686377	Platinum derivatives	Impaired response	[133]
	*IDH1*	rs121913499	Ivosidenib	Increased response	[134]
		rs121913500			

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
