# Peer review of "Role of Genetic Variations in the Hepatic Handling of Drugs"

_ijms, 2020, doi:10.3390/ijms21082884_

Round 1

Reviewer 1 Report

In their manuscript “Role of genetic variations in the hepatic handling of drugs”, Marin, et. al. provide an expansive overview of genetic polymorphisms that may impact drug disposition or effect, including genes encoding transporters, drug metabolizing enzymes, and drug targets. I believe this topic would be of interest to the readership of “International Journal of Molecular Sciences”. However, there are a number of issues with the manuscript that need to be addressed prior to publication. I have provided specific comments below.

  • The information provided under each subheading appears inconsistent. Given the topic of the paper, it seems that critical information for each gene or set of genes would include: frequencies of well-established variants, functional effects of variants (e.g., protein truncating, lower expression, lower activity, etc.), demonstrated effects on clinical PK, PD/efficacy, and safety for affected drugs, and potential clinical impact (e.g., whether or not dose changes or alternative agents should be considered). However, most sections are missing at least some of this information. For example, the section on COMTs lists a genetic variant and states it is reduced function, and also lists classes of substrates of COMT, but does not state whether or not any non-clinical or clinical data support an impact of the variant on any drugs. I understand that not all of this information will be available for each gene or variant, but a comprehensive review should have the majority of this information, or state that data are not available or inconclusive.
  • Table 1 is potentially confusing and misleading. Not all of the variants listed have been shown to affect all of the drugs listed or to result in all of the consequences listed. The table should either provide more comprehensive information, including supporting references, or make it clear that these are representative variants, drugs, and consequences and not all combinations of variants, drugs, and consequences are supported.
  • The length of the manuscript makes it impractical to verify every statement and reference; however, in reading through it I identified a number of errors, including:
    • Line 63: The FDA does not have any official recognition of the PharmGKB database. The ClinGen resource is recognized by the FDA’s database recognition program, but only for germline variants for hereditary diseases, not the variants described within the manuscript.
    • Line 165: CYP2C19*17 is listed as a loss-of-function allele, it is actually a gain-of-function allele.
    • Lines 208-210: The manuscript states “There are up to 80 allelic variants but only 5 of them determine the phenotype IMs (*9, *10, *17, *29 and *41), 3 that of PMs (3 to *8 and *36) and 3 that of UMs (*1xN, *2, and *35) [47].” However, the reference provided does not seem to support the limited number of alleles contributing to the listed phenotypes, and CPIC lists substantially more alleles that can contribute to the phenotypes. In addition, the reference states there are over 100 cataloged star variants while the paper states only 80.
  • There are several statements about associations between genetic variants and disease risk, which seems beyond the scope of the paper as outlined by the title.

Author Response

Reviewer #1

COMMENT

In their manuscript “Role of genetic variations in the hepatic handling of drugs”, Marin, et. al. provide an expansive overview of genetic polymorphisms that may impact drug disposition or effect, including genes encoding transporters, drug metabolizing enzymes, and drug targets. I believe this topic would be of interest to the readership of “International Journal of Molecular Sciences”. However, there are a number of issues with the manuscript that need to be addressed prior to publication. I have provided specific comments below.

The information provided under each subheading appears inconsistent. Given the topic of the paper, it seems that critical information for each gene or set of genes would include: frequencies of well-established variants, functional effects of variants (e.g., protein truncating, lower expression, lower activity, etc.), demonstrated effects on clinical PK, PD/efficacy, and safety for affected drugs, and potential clinical impact (e.g., whether or not dose changes or alternative agents should be considered). However, most sections are missing at least some of this information. For example, the section on COMTs lists a genetic variant and states it is reduced function, and also lists classes of substrates of COMT, but does not state whether or not any non-clinical or clinical data support an impact of the variant on any drugs. I understand that not all of this information will be available for each gene or variant, but a comprehensive review should have the majority of this information, or state that data are not available or inconclusive.

RESPONSE

As suggested by the reviewer, the allelic frequency of the most clinically relevant variants in the hepatic handling of drugs or affecting drugs targeting the liver has been included. In addition, we have tried to ensure that for all these variants, the information on functional effects, when known, the effect on pharmacokinetics, efficacy or toxicity, according to demonstrated studies that are collected in the various databases, and recommendations on changes in dosage or substitution of drugs has been included.

However, it should be considered that this review does not intend to be an exhaustive study gathering all the existing data on all the variants and the drugs that it affects. Our intention is to offer the reader a general idea of which variants are really relevant from the clinical point of view and could be crucial for the development of modern personalized pharmacology aimed at increasing the effectiveness of the drug and/or mitigating its adverse reactions in each patient. Based on the reviewer’s comment this focus has been improved in this new version of the manuscript.

COMMENT

Table 1 is potentially confusing and misleading. Not all of the variants listed have been shown to affect all of the drugs listed or to result in all of the consequences listed. The table should either provide more comprehensive information, including supporting references, or make it clear that these are representative variants, drugs, and consequences and not all combinations of variants, drugs, and consequences are supported.

RESPONSE

In the revised version of the manuscript, we have modified the tables by including only the genetic variants that have the greatest clinical relevance and the drugs that they affect, as recorded in different databases and clinical guidelines. In addition, we have added references from the studies that support the data included in the tables.

COMMENT

The length of the manuscript makes it impractical to verify every statement and reference; however, in reading through it I identified a number of errors, including:

Line 63: The FDA does not have any official recognition of the PharmGKB database. The ClinGen resource is recognized by the FDA’s database recognition program, but only for germline variants for hereditary diseases, not the variants described within the manuscript.

RESPONSE

This sentence has been corrected.

COMMENT

Line 165: CYP2C19*17 is listed as a loss-of-function allele, it is actually a gain-of-function allele.

RESPONSE

The reviewer is right. There was a typing error as later described on lines 173 and 174. We have removed it from the text.

COMMENT

Lines 208-210: The manuscript states “There are up to 80 allelic variants but only 5 of them determine the phenotype IMs (*9, *10, *17, *29 and *41), 3 that of PMs (3 to *8 and *36) and 3 that of UMs (*1xN, *2, and *35) [47].” However, the reference provided does not seem to support the limited number of alleles contributing to the listed phenotypes, and CPIC lists substantially more alleles that can contribute to the phenotypes. In addition, the reference states there are over 100 cataloged star variants while the paper states only 80.

RESPONSE

The reviewer is also right here. We have already corrected the statement on lines 249-250.

COMMENT

There are several statements about associations between genetic variants and disease risk, which seems beyond the scope of the paper as outlined by the title.

RESPONSE

We agree with reviewer on this criticism that weakened the whole message in the previous version of manuscript. This has been changed by removing all comments and references regarding any association between genetic variants and disease risk.

Reviewer 2 Report

Review presents an in-depth analysis of Role of Genetic Variations in the Hepatic Handling of Drugs. Manuscript is well written, easy to follow and well organized. Information is presented at a sufficient level of detail even as authors aimed to cover a very large scope of research. I had no substantial comments. However I recommend authors to proof read for minor grammatical errors throughout the manuscript.  

In abstract , last sentence is very confusing. I request authors to rephrase this sentence to convey correct meaning. Also at several places symbol "=" is included and also highlighted with yellow color in the text. Purpose of this symbol is not clear. So authors should correct this.

Author Response

Reviewer #2

COMMENT

Review presents an in-depth analysis of Role of Genetic Variations in the Hepatic Handling of Drugs. Manuscript is well written, easy to follow and well organized. Information is presented at a sufficient level of detail even as authors aimed to cover a very large scope of research. I had no substantial comments. However I recommend authors to proof read for minor grammatical errors throughout the manuscript.  

In abstract , last sentence is very confusing. I request authors to rephrase this sentence to convey correct meaning. Also at several places symbol "=" is included and also highlighted with yellow color in the text. Purpose of this symbol is not clear. So authors should correct this.

RESPONSE

We appreciate the positive evaluation of the reviewer. The abstract has been changed to improve the clarity of the last sentence.

The symbol "=" is commonly used in mutation nomenclature to indicate that the amino acid has not been changed. Based on the reviewer comment this has been omitted through the text. The yellow highlighting was an editing mistake. 

Round 2

Reviewer 1 Report

The authors have extensively edited the manuscript and I find it much more cohesive, easier to follow, and accurate. I believe that the manuscript is suitable for publication and will be an excellent contribution to the literature, provided that the minor edits suggested below are addressed.

  • The terms mutation and variant/SNP are used interchangeably, which is sometimes inappropriate. For example, the SLCO1B1 521T>C polymorphism is referred to as a mutation. However, given that this is a common germline genetic variant, it should be referred to as a variant or SNP. The authors should correct this terminology throughout. In addition, since the majority of the paper focuses on germline variants, the authors should specify any somatic mutations that are discussed in the section on cancer therapeutics.
  • Lines 103-105: The statement about clinical guideline recommendations should be referenced.
  • Table 1: CYP2C19 - clopidogrel is a prodrug and has lower plasma concentrations of the active metabolite and lower activity. The PPIs have higher plasma concentrations but not lower activity. This should be clarified.
  • Table 1: CYP2C8 - The statement of enhanced toxicity seems contradictory to the statement on lines 207-208.
  • Lines 217-219: "The most studied polymorphisms are *2 and *3, because all carriers experienced hypoglycemia during treatment for type 2 diabetes mellitus, and bleeding when treated with NSAIDs [63]." This statement is not accurate the referenced paper indicates that carriers are "at risk" not that all experienced AEs.
  • Lines 236-238: Although the authors corrected the number of variants, the paper still states that a minimal number of variants contribute to each phenotype, which does not seem accurate based on the CPIC reference.
  • Lines 252-259: Although CYP3A5 genotype has not been associated with outcomes, tacrolimus dosing is guided by therapeutic drug monitoring, and CYP3A5 genetics are associated with lower systemic concentrations and lower probability of achieving target concentrations in patients treated with tacrolimus (in addition to being incorporated into dosing algorithms as the authors indicate). Please consider adding this additional information (references can be found in the CPIC guideline).
  • Line 359: The word "profitable" does not seem appropriate. Suggest "useful" or "important".

Author Response

Manuscript ID: ijms-742774-R2

Reviewer #1

COMMENT

The authors have extensively edited the manuscript and I find it much more cohesive, easier to follow, and accurate. I believe that the manuscript is suitable for publication and will be an excellent contribution to the literature, provided that the minor edits suggested below are addressed.

RESPONSE

We would like to express our sincere gratitude for the positive evaluation and for the profound reading of our manuscript, which will permit us to improve the text by including all suggested changes.

COMMENT

The terms mutation and variant/SNP are used interchangeably, which is sometimes inappropriate. For example, the SLCO1B1 521T>C polymorphism is referred to as a mutation. However, given that this is a common germline genetic variant, it should be referred to as a variant or SNP. The authors should correct this terminology throughout. In addition, since the majority of the paper focuses on germline variants, the authors should specify any somatic mutations that are discussed in the section on cancer therapeutics.

RESPONSE

The reviewer is right. This is an important point that has been addressed to improve the accuracy of the text. The terms SNP/variant/polymorphism and mutation had sometimes been used with little precision. Each of the described variants and mutations has been reviewed and those that were misnamed have been corrected. Moreover, we agree with the reviewer that the somatic mutations discussed in the section on cancer therapeutics were not specified. We have pointed out those mutations either directly, referring to them as somatic mutations, or indicating that they have been identified in cancer tissues.

COMMENT

Lines 103-105: The statement about clinical guideline recommendations should be referenced.

RESPONSE

This improvement has been done. A reference has been provided and it has been specified which clinical guidelines published this recommendation to change the drug treatment.

COMMENT

Table 1: CYP2C19 - clopidogrel is a prodrug and has lower plasma concentrations of the active metabolite and lower activity. The PPIs have higher plasma concentrations but not lower activity. This should be clarified.

RESPONSE

The reviewer is right. This issue has been corrected. We have subdivided the CYP2C19 section in Table 1 to separate these two types of drugs to prevent any confusion.

COMMENT

Table 1: CYP2C8 - The statement of enhanced toxicity seems contradictory to the statement on lines 207-208.

RESPONSE

We agree with the reviewer’s comment. This apparently contradictory statement has been clarified. We have deleted “enhanced toxicity” in Table 1.

COMMENT

Lines 217-219: "The most studied polymorphisms are *2 and *3, because all carriers experienced hypoglycemia during treatment for type 2 diabetes mellitus, and bleeding when treated with NSAIDs [63]." This statement is not accurate the referenced paper indicates that carriers are "at risk" not that all experienced AEs.

RESPONSE

The reviewer is right. This misleading statement has been corrected.

COMMENT

Lines 236-238: Although the authors corrected the number of variants, the paper still states that a minimal number of variants contribute to each phenotype, which does not seem accurate based on the CPIC reference.

RESPONSE

This paragraph has been changed to improve its accuracy based on the available bibliography.

COMMENT

Lines 252-259: Although CYP3A5 genotype has not been associated with outcomes, tacrolimus dosing is guided by therapeutic drug monitoring, and CYP3A5 genetics are associated with lower systemic concentrations and lower probability of achieving target concentrations in patients treated with tacrolimus (in addition to being incorporated into dosing algorithms as the authors indicate). Please consider adding this additional information (references can be found in the CPIC guideline).

RESPONSE

Thank you for the recommendation. This information has been added together with a new reference.

COMMENT

Line 359: The word "profitable" does not seem appropriate. Suggest "useful" or "important".

RESPONSE

This word has been changed as suggested by the reviewer.